# Hash and Physical Unclonable Function (PUF)-Based Mutual Authentication Mechanism

**DOI:** 10.3390/s23146307

**Published:** 2023-07-11

**Authors:** Kavita Bhatia, Santosh K. Pandey, Vivek K. Singh, Deena Nath Gupta

**Affiliations:** 1Department of Computer Science, Banaras Hindu University, Varanasi 221005, India; kbhatia@gov.in (K.B.); vivek@bhu.ac.in (V.K.S.); 2Ministry of Electronics and IT, Government of India, New Delhi 110003, India; santosh.pandey@meity.gov.in; 3Center for Development of Advanced Computing, Mumbai 400049, India

**Keywords:** hash, PUF, broken authentication, mutual authentication, privacy-preserving protocol

## Abstract

The security of web applications in an enterprise is of paramount importance. To strengthen the security of applications, the identification and mitigation of vulnerabilities through appropriate countermeasures becomes imperative. The Open Web Application Security Project (OWASP) Top 10 API Security Risks, 2023 Edition, indicates the prominent vulnerabilities of API security risks. Broken authentication, however, is placed in second position with level-3 exploitability, level-2 prevalence, level-3 detectability, and level-3 technical impact. To mitigate this vulnerability, many mitigation strategies have been proposed by using the cryptographic primitives wherein two techniques, namely hashing and PUF, are used. Some of the proposals have integrated the concepts of hashing and PUF. However, the unnecessarily lengthy and complex mathematics used in these proposals makes them unsuitable for current API-based application scenarios. Therefore, in this paper, the authors propose a privacy-preserving authentication protocol that incorporates the capability of both mechanisms in an easy and low-complexity manner. In addition to overcoming existing limitations, the proposed protocol is tested to provide more security properties over existing schemes. Analysis of their performance has demonstrated that the proposed solutions are secure, efficient, practical, and effective for API-based web applications in an enterprise environment.

## 1. Introduction

The ultimate goal of security providers is to make their enterprises free from all types of vulnerabilities. To achieve this goal, security providers are working continuously to identify the current threats to the security systems [1,2]. One such identified threat is unauthorized authentication to the system. In this, unauthorized users can get access to restricted resources of the enterprise. The security flaws identified for this type of attack are termed “Broken Authentication” vulnerabilities, which basically result in weak authentication and authorization. These allow attackers to gain access to sensitive data of the enterprise by circumventing security procedures [3]. Broken authentication vulnerabilities can be exploited in many ways, including injection flaws, cross-site scripting (XSS), authentication problems, and session management problems. In one case of broken authentication, a regular user might get the rights to add new users. This will pose a serious security threat to the system because the authority to add new users must be limited to the administrator only [4,5]. Seeing the current scenario of broken authentication vulnerabilities, it is crucial to implement security measures such as input validation, session management, and authorization controls [6].

Hashing and random nonce values can be used to strengthen the security mechanisms for such systems. Additionally, they must be revisited before their use in current broken authentication vulnerability scenarios. Hashing has now advanced to sponge construction. The sponge-based hash constructions are claimed to be resistant up to second-preimage attacks. Sponge construction is used to create a hash function that maps variable-length input to variable-length output through a fixed-length permutation (or transformation) and padding rules. The function accepts arbitrary binary strings or elements of Z_2_ as input and returns a binary string or element of (Z_2_)^n^ based on the value of n entered by the user. In the construction of a sponge, there are three phases: the initialization, absorption, and squeezing phases. It utilizes the states of b = r + c bits in its construction [7].

The bitrate, b, is the sum of the rate, r, and the capacity, c. The input string is padded with a reversible padding rule after being divided into blocks of r bits. A sponge is then constructed in two stages, using a state whose b bits have been set to zero. It is first necessary to XOR the input blocks’ r bits into the initial r bits of the state, with f applications interspersed between them during the absorption phase. Following the processing of each input block, the sponge architecture enters the squeezing phase. In the second step, the first r bits of the state are output blocks that are separated by the function, f, during the squeezing phase. There is no restriction on how many chunks of output should be generated. When the squeezing is performed, the final c bits of the state are not emitted and are not directly affected by the input blocks [8].

In addition to hashing, PUFs are also becoming increasingly popular as a security primitive for communicating devices. This is a function that produces its output on the basis of a random nonce and the hardware details of the device. A PUF is produced during the manufacturing process of an integrated circuit (IC); its uniqueness is a result of arbitrary physical verification of the IC microstructure [9]. The PUF is basically an integrated circuit (IC) that provides a one-way function that cannot be duplicated, due to its internal structure. It is easy to make the PUFs with a few gates; they are difficult to predict, but again it is easy to evaluate their outputs. These characteristics make them suitable for use as security primitives for communication devices. The intrinsic random physical nature of PUFs and their uncontrollability, as well as their inherent random behavior (the challenge–response pair or CRP), makes it hard to predict and replicate their behavior [10,11].

In global manufacturing chains, PUF can be used to track, identify, and authenticate hardware with low overhead. It has been more than a decade since the first PUF was introduced [12]. The silicon PUF has been implemented in many ways since then, including the Arbiter PUF [10], Ring Oscillator (RO) PUF [13], static random access memory (SRAM) PUF [14], and many others. A silicon PUF designer must address two of the most challenging design tasks a PUF designer will encounter, based on the fact that all current silicon PUF implementations use parametric manufacturing variations: What are the best ways to make the PUF extremely stable and unique, even when exposed to extreme conditions, without high implementation costs? Given the uniqueness and stability measurements of a PUF, how can security be evaluated?

Although many works have already been proposed using these mechanisms, utilizing both simultaneously for API-based web applications has not been seen anywhere. In the presented paper, the authors propose a novel concept of mutual authentication. The simplest form of mutual authentication involves handshakes and information exchange between sender and receiver. Through this process, both entities are confirmed to be trustworthy and to be who they claim to be. As a result of mutual authentication, there is less risk that a network user might inadvertently reveal sensitive information to a malicious or insecure website. Considering the zero-trust approach towards transmission security, the authors in this paper propose a shield handshake between the client and the server by hashing the credentials.

## 2. Related Work

In recent years, many interesting anonymous authentication schemes have been proposed that preserve privacy. They can be divided into three categories: (1) public key cryptosystems (PKC), (2) error correction systems (EC), and (3) symmetric key cryptosystems. In most PKC-based schemes [15,16,17,18,19,20], the cryptosystems are elliptic curves, which have an extremely high hardware cost, making them unfeasible to implement for most clients. The error rate of EC-based schemes [21,22,23,24,25] has to be below a certain threshold value in order for them to prove secure. Moreover, these schemes are not scalable, as they are only capable of supporting a limited number of clients. In general, NPKC schemes fall into two categories: schemes based on hashes [26,27,28,29] and schemes based on PUFs [30,31,32,33,34,35]. Any physical or cloning attacks cannot be prevented using hash-based schemes. This has led to the growing popularity of PUF-based schemes. Mutual authentication systems have been implemented using PUFs in a number of studies.

A tree-based authentication scheme using PUFs was proposed by Bringer et al. in 2008 [30]. This protocol, however, does not provide protection against DoS attacks and impersonation attacks [31]. Subsequently, Sadeghi et al. [31] proposed an additional PUF-based scheme, but Kardas et al. discovered that it is vulnerable to cold boot attacks [33]. Once the scheme has been compromised, an attacker will be able to impersonate it easily and even trace its past and future communications. Another tree-based authentication protocol based on PUFs was presented by Akgun and Caglayan [32]. Their protocol, however, is also susceptible to the cold boot attack presented in [33], which means that they are unable to guarantee the desired security properties. A new PUF-based authentication protocol was also proposed by Kardas et al. [34] in 2012, but the scheme cannot guarantee forward secrecy and is not resilient to DDoS attacks. The authorization scheme proposed by Jung et al. uses PUF for HMAC-based authentication, yet it is also susceptible to DoS attacks.

An additional PUF-based authentication scheme for communicating systems was recently proposed by Akgun and Caglayan [35]. Investigation of the scheme, however, revealed that forward secrecy is not supported, which is a necessary security requirement for communicating systems. Additionally, the authors found that some PUF-based schemes were impractical, which resulted in servers performing exhaustive searches in order to identify clients, negatively affecting performance. A hash function is used, for example, to encode the identity of the client in [35]. As a result, the backend server would need to try all possible combinations of secrets to determine the client. Other PUF-based authentication schemes also exhibit a similar scenario. The secret keys must also be stored in the client’s memory in all the existing schemes. Once the client is compromised, aside from the storage costs, the attacker has access to those secrets and can perform several attacks.

Assertions of noise-resilient or ideal PUFs have been used in all the aforementioned schemes [36,37]. For this proposal, as well, the authors are assuming that the PUFs are ideal and noise-resilient. In this paper, the authors propose an improved mutual authentication mechanism based on hashing and PUF. Similar work has already been published by Bendavid et al. [38], Xu et al. [39], and Kulseng et al. [40]. However, Bendavid and Xu’s works are largely based on the proposal made by Kulseng. So, the authors in this paper studied the mechanism proposed by Kulseng and identified some of the gaps that could be used by an attacker to get access to the system. A detailed analysis of Kulseng’s proposal is explained in the next section.

## 3. Kulseng’s Mutual Authentication Protocol

Lars Skaar Kulseng proposed a lightweight mutual authentication, an owner transfer, and a secure search protocol in 2009 [40]. The mechanism starts with the sending of a search request from the server to the client. The client will respond with its IDS (index of the client in the database (DB)). The server will find the ID of the client from its DB with the help of the IDS sent by the client. The server then computes the ID⊕G_n_ and sends it to the client. The client, by using its ID, can find the G_n_ of the server by calculating ID⊕G_n_⊕ID. On the basis of this G_n_, the client calculates four variables, namely, G_n+1_, G_n+2_, K_n_, and K_n+1_. The client sends two different values to the server, namely (K_n_⊕G_n+1_) and (K_n+1_⊕G_n+2_). The values of G_n+1_ and G_n+2_ will be calculated on the basis of the PUF, whereas K_n_ and K_n+1_ will be calculated on the simple function, F. Also, the client updates its IDS by the following operation: IDS = F(IDS⊕G_n_), and the value of G_n+1_ will be stored in place of Gn. The server will do a cross-check of the values. After a successful match, it will update the values by the following operations: G_n+1_ = (K_n_⊕G_n+1_)⊕F(G_n_), G_n+2_ = (K_n+1_⊕G_n+2_)⊕F(F(G_n_)), and IDS = F(IDS⊕G_n_). The value of G_n+1_ will be stored in place of Gn, and the value of G_n+2_ will be stored in place of G_n+1_. The server can only use the simple function, F, for all its operations because the PUF is associated with the client only.

The above-mentioned procedure provides three things: client verification, server verification, and mutual authentication between the client and the server. Also, the calculated values of different functions, namely IDS, G_n+1_, and G_n+2_, are in line for both the client and the server to be used in the next communication between the same client and the server. However, in 2018, Xu et al. [39] found some serious security loopholes in the mutual authentication process of Kulseng’s design. The claims raised in [39] are as follows. From the communication channel, the interceptor will steal (ID⊕G_n_), (K_n_⊕G_n+1_), and (K_n+1_⊕G_n+2_) in the first communication setup and the IDS from the next communication setup. Because the IDS used in subsequent communication is calculated by IDS = F(IDS⊕G_n_), Xu et al. claim that the G_n_ of the server can be found by just XORing the new IDS with the F(IDS⊕G_n_). However, this seems not to be true. Two questions arise from here. First, when the update process of the IDS is happening inside the client and server individually, then how can an intruder get it? Second, when the new IDS is a function of (IDS⊕G_n_), and not just (IDS⊕G_n_), then how can an intruder find the value of G_n_ by just XORing it with the old IDS? Also, the authors in [39] claim that the seeds of an LFSR can be predicted by using the polynomial used and the last output value generated by this LFSR. If this really happens, then the optimal proposal of security may advocate the use of NLFSR in place of LFSR.

In the second claim, the authors in [39] state that the intruder will receive the information sent by the client, and it will never reach the server. The authors in [39] try to convince the readers that, in this way, the DB will not be updated and the verification of the client will fail the next time. If this happens, the optimal solution should enable a negative acknowledgment in communication between the client and the server. The server must send a negative acknowledgment after a specific amount of time to the client about its non-receipt of the information. Also, the client should wait for a specific amount of time for the acknowledgment from the server. The proposed solutions from [39] do not contain any of the above-mentioned techniques; rather, they propose a very complex procedure for client verification, server verification, and mutual authentication between client and server. Their proposed solution not only forfeits the requirement of the devices under a constrained environment but also is subject to mathematical verifications. The authors of [39] also accept the drawback related to storage in their design. The database needs to store nine different values about a single Tag all the time. Also, these values keep changing after every communication. This shows the non-suitability of the proposal made by [39] in current communicating environments.

In the client verification phase, the authors in [39] stated that the client will use r_1_ and FID^new^ for verification, if not intercepted, and r_1_ and FID^old^ if intercepted. The obvious question here is: Do the clients know whether they have been intercepted or not? Another security loophole is: Can one not intercept the r_1_ and FID^new^ in between to extract the value of FID^new^ or r_1_ and FID^old^ to extract the value of FID^old^, whichever is applicable? In the mutual verification phase, the calculated value of the client’s r_2′_ is P_n+1_⊕r_2_⊕D. D is calculated by the client as PUF(P_n_). Now, it cannot be proven that PUF(P_n_) is equal to P_n+1_. Hence, B = B′ also cannot be proven. The process of server verification is suspicious. Again, in the client verification phase, a doubtful calculation is shown. The server is calculating the value of F′ on the basis of the received value of F from the client. This will always produce F′ = F. Hence, it cannot be used in the verification process. This means that H = H′ cannot be proven. Another obvious question arises from this point: when the client has been verified at the very first step, then why does the protocol verify the client again at the later stage? Also, the authors of [39] do not provide the area and power required for their algorithm to run.

## 4. Proposed Mutual Authentication Mechanism

The proposed mechanism consists of three phases: server verification, client verification, and mutual authentication. The definitions of different symbols used in the proposed protocol are given in Table 1.

### 4.1. Server Verification

Each client, at the time registration, is provided a hashed identification of their corresponding server. On every new entry to the environment, the server broadcasts a search request. It sends an encrypted message to the client. The client, before responding to that message, runs the server verification code. The client responds to the server only after a successful verification of the server. The protocol for server verification goes as follows:

Algo: Server verification

The server performs hashing over its identity H(S_ID_) and attaches the current timestamp, T_1,_ to it. Then, it encrypts all the data with a symmetric key shared with the client at the time of registration.The server then sends the first message to the client: M_1_ = E[H(S_ID_)||T_1_].The client then decrypts the whole message, M1, by using the symmetric key given to it at the time of registration. It gets the hashed identification of the server by dropping the timestamp part of the received message.The client compares the received H(S_ID_) with the one it already has.If it is a match, the server is verified; otherwise, it is not verified.

### 4.2. Client Verification

After the successful server verification, the client performs the XOR operation on its own identification, i.e., C_ID_, with the server’s hashed identification, i.e., H(S_ID_). The client then appends a new timestamp, T_2_, to the entire message. It then encrypts the whole message with the server’s public key. This encrypted message is then sent to the server. The protocol for client verification goes as follows:

Algo: Client verification

The client performs the XOR operation on its own identity, C_ID,_ along with the hashed identity of the server, H(S_ID_). It then attaches a new timestamp, T_2,_ to this message. The client encrypts the whole message with the server’s public key, which is broadcast to all the clients on a regular basis.The client then sends a second message to the server: M_2_ = E[(H(S_ID_)⊕C_ID_)||T_2_].At the server end, the first check is on the timestamp. It will check if the current timestamp is greater than the one the server sent to the client earlier: T_2_ > T_1_.The server then decrypts the whole message by using its private key.It drops the timestamp part of M_2_ to get the message part. The server then extracts the client identification, C_ID_, from (C_ID_⊕H(S_ID_)) by XORing it with its own hashed identity, H(S_ID_).The server asks the DB about C_ID_. If there is a match, the client is verified; otherwise, it is not verified.

### 4.3. Mutual Authentication

After the successful verification of the server and the client, a PRNG unit will be activated to generate a secret value, G_n_, at the server end. The server then applies a simple function over the generated value to get F(G_n_). It again repeats the process to get another value, F(F(G_n_)). In a separate computation, the server XORs the client’s identity, C_ID,_ with the generated value, G_n_, and then attaches a new timestamp, T_3_, to it. The server encrypts the entire message with the client’s public key. The server then sends the third message, M_3_: E[(C_ID_⊕G_n_)||T_3_], to the client. At the client end, the first check will be on the timestamp. If the current timestamp will be greater than the old timestamp, (T_3_ > T_2_), then only the client will decrypt the whole message using its own private key. It then drops the timestamp part to get the remaining message. By XORing the received message with its own identity, the client will receive the value of G_n_.

After receiving the value of G_n_, the client will also apply a simple function over it to get the value for F(G_n_). The client repeats the process to get another value, F(F(G_n_)). The client also updates the value of G_n_ to G_n+1_ by applying a special PUF function to G_n_ and then to G_n+2_ by applying the same PUF function to G_n+1_. The client then performs two separate XORs over two different sets of values; in the first set, it performs XOR over G_n+1_, K_n_, and C_ID_, and, in the second set, it performs XOR over G_n+2_, K_n+1_, and C_ID_. The client then appends a new timestamp, T_4_, to both values and encrypts both messages with the server’s public key. The client sends messages M_4_ and M_5_ to the server: M_4_: E[(G_n+1_⊕K_n_⊕C_ID_)||T_4_] and M_5_: E[(G_n+2_⊕K_n+1_⊕C_ID_)||T_4_]. The server first checks the value of the received timestamp. If the current timestamp, T_4_, is greater than the previous timestamp, T_3_, only then will it proceed further.

The server will decrypt both messages by applying its private key. It drops the timestamp field to get the original message. The server then performs XOR with the customer identification, C_ID_, over the received message, and then performs a second XOR with K_n_ and K_n+1_ separately to get the values of G_n+1_ and G_n+2_. After successful mutual authentication, both parties update the client identification and generator values for future communication. The client updates its identification by XORing its old identification with the generator value and applying a simple function over it: C′_ID_: F(C_ID_⊕G_n_). The same is done at the server end. Both the client and the server save the updated set of values [G_n+1_, G_n+2_, and C′_ID_]. In this set, the value of G_n+1_ will act as G_n_, and the value of G_n+2_ will act as G_n+1_ for further communication.

Algo: Mutual authentication

The server activates the PRNG unit to generate the value, G_n_.The server performs an XOR between C_ID_ and G_n_, then appends a new timestamp, T_3_, and encrypts the entire message with the client’s public key.It sends message M_3_ to the client: M_3_: E[(C_ID_⊕G_n_)||T_3_].The client first checks the timestamp. If T_3_ > T_2_ only then does it decrypt the entire message using its own private key.The client drops the timestamp field to get the original message. It then extracts the value of G_n_ by performing (C_ID_⊕G_n_)⊕C_ID_.The server computes F(G_n_) and F(F(G_n_)), while the client simultaneously computes G_n+1_ = PUF(G_n_), G_n+2_ = PUF(G_n+1_), K_n_ = F(G_n_), and K_n+1_ = F(F(G_n_)).The client performs the XOR operation between G_n+1_ and K_n_ and C_ID_: G_n+1_⊕K_n_⊕C_ID_.The client performs a second XOR operation between G_n+2_ and K_n+1_ and C_ID_: G_n+2_⊕K_n+1_⊕C_ID_.The client then appends a new timestamp, T_4_, to both messages.It then encrypts them with the server’s public key: E[(G_n+1_⊕K_n_⊕C_ID_)||T_4_], E[(G_n+2_⊕K_n+1_⊕C_ID_)||T_4_].The client sends M_4_ and M_5_ to the server.The server first checks the new timestamp. If T_4_ > T_3,_ only then will the server decrypt both messages. It then drops the timestamp part to get the original message. The server extracts the values of G_n+1_ and G_n+2_ by the following operations: (G_n+1_⊕K_n_⊕C_ID_)⊕F(G_n_⊕C_ID_) and (G_n+2_⊕K_n+1_⊕C_ID_)⊕F(F(G_n_)⊕C_ID_).The client and server simultaneously update the client identification, C′_ID,_ by F(C_ID_⊕G_n_).They both store the set of new parameters as: G_n+1_, G_n+2_, and C′_ID_.

The time-sequence diagram for the entire mechanism is presented in Figure 1.

## 5. System Model

A distributed information system, in particular a client-server architecture, has been developed as a result of modern advances in information technology and high-load computing systems. An “information system with client-server architecture” refers to one where functional components interact in a “request–response” manner [41,42]. In such systems, servers usually contain the database and software, while the client components usually include corresponding hardware and software parts, as well as client interfaces. It is most common for data processing to be shared between client and server components. Requests are initiated by the client, and responses are generated by the server. There is an important need for developing models of distributed information systems in stationary conditions and under the influence of external factors that increase the failure rate of component systems [43,44].

## 6. Security Evaluation of the Proposed Mechanism

Formal and informal evaluations of the security of the proposed mechanism are presented in this section. The discussion focused on an active adversary, who can eavesdrop on any messages transferred over a public channel at different stages of the process, such as during the authentication phase of the protocol, by modifying/blocking messages, or by initiating a session to impersonate a server/client. Nevertheless, the authors specify that the adversary cannot access or influence transmitted messages over a secured channel, such as the registration phase. Random behavior is also assumed for the embedded PUF when different inputs are used.

### 6.1. Informal Analysis

#### 6.1.1. Traceability Attack

It is necessary for an adversary to link a client’s identity or previous messages transferred to a server to trace a client and compromise the protocol’s anonymity. In the proposed mechanism, the hashed identities (H(S_ID_)) are shared on the communication channel. Hence, it is almost impossible for an adversary to get back the original identity of the client/server if it manages to discover the hashed identity of the client/server. Messages M_1_ and M_2_ can be seen for this reference. Additionally, after each run of the protocol, the value of the client identity is updated to C’_D_. So, it is not possible to trace the identity of a client from its previous communication. Hence, the authors conclude that the proposed mechanism is free from traceability attacks.

#### 6.1.2. Secret Disclosure Attack

In the proposed mechanism, the client is sharing only three messages (M_2_, M_4_, and M_5_) with the server. M_2_ is an encryption over XOR between H(S_ID_) and C_ID_ attached with a timestamp, M_4_ is an encryption over XOR between G_n+1_, K_n_, and C_ID_ attached with a timestamp, and M_5_ is an encryption over XOR between G_n+2,_ K_n+1_, and C_ID_. An adversary cannot invert the value of (H(S_ID_)), and, hence, it cannot discover the value of either S_ID_ or C_ID_ from M_2_. In the same way, the values of K_n_ and K_n+1_ cannot be extracted from the messages M_4_ and M_5_. Hence, the authors conclude that the proposed mechanism is free from a secret disclosure attack.

#### 6.1.3. Impersonation Attack

To impersonate a client on the server without determining C_ID_ and S_ID_, the adversary must generate at least a valid M_3_. For the adversary to produce M_3_, the knowledge of C_ID_, however, is essential, since the eavesdropped messages cannot be utilized because each new session generates a fresh G_n_, making this attack inefficient. Similar arguments apply to client impersonation attacks on the server.

#### 6.1.4. DDoS Attack

It is not possible to apply a DDoS attack against the proposed mechanism, since all messages that are critical to verify the client’s authentication are randomized by Gn. If a cloned client wants to apply a DDoS attack, it must be able to be recognized by the server first. Since the first message sent by the server is protected with a symmetric key known only to the registered clients, no outsider can have the same key to decrypt the message from the server. In case the adversary somehow succeeds in arranging the correct key, it will not be able to recognize the correct timestamp pattern. In case the adversary is able to recognize the correct pattern, it will not have the hashed identity of the server. So, we can say, without having the server’s identification, the cloned client cannot produce a successful denial of service attack.

#### 6.1.5. Desynchronization Attack

Since an adversary cannot impersonate the server to the client or the client to the server in order to desynchronize the client and the server, another approach is to block the last message of the current communication. In this case, the updating of the records will only take place at the client end but not at the server end. The desynchronization attack will not have any effect on the proposed mechanism because, in this case, both parties agreed on an old data set to store. If one party is not updated with a new set of data, both will communicate on the old set of data.

#### 6.1.6. Forward/Backward Secrecy

Both NLFSR and PUF are computed on the basis of a generator value, Gn. This generator value is the product of a PRNG [45] working separately. The PRNG produces perfectly random nonce values each time it is called and, hence, cannot be determined by an adversary for the next time, no matter whether it is known for all the previous sessions. Furthermore, the server sends M_3_ to the client, where M_3_ is an XOR of the client’s ID and PRNG output. Since the PRNG output changes every time, it is impossible for an adversary to break the message, M_3,_ from the server to the client. Similarly, messages M_4_ and M_5_ are also unbreakable.

### 6.2. Formal Analysis

An open-source tool called Scyther has become widely accepted for evaluating protocol security. Using it, users can analyze complex attack scenarios on the target protocol using a graphical user interface. With Scyther, the authors verified whether their assertions about security held for the protocol. As part of the Scyther process, the authors defined two identities, namely, the client and the server. The implementation details are presented in Appendix A, where we describe each role and its corresponding task. The authors also fixed the number of protocol runs to 100, the search pruning to “Find all attacks”, and the matching type of the Scyther tool to “Find all type flaws”. All the messages sent and received were covered by the proposed secret claims. The authors established that the security allegations are well-founded in Table 2, showing that the Scyther tool was unable to find any attacks within bounds. Additionally, we compared the improved protocol with some other protocols in Table 3 to determine its security level. The results show that the proposed protocol is secure and resistant to advanced attacks, addressing the security problems of other protocols.

## 7. Complexity Analysis

In the client verification phase, a timestamp is generated in the client recognition phase, so the time complexity of the client is Tc. The time complexity of an XOR operation is T_XOR_, and the time complexity for the encryption is T_ENC_. In the process of authenticating the server in the mutual verification phase, there are four operations by a client, namely decryption, drop, XOR, and comparison. So, in the process of authenticating the server in the mutual verification phase, the time complexity of the client is (Tc + T_ENC_ + T_XOR_). In the process of authenticating the client in the mutual verification phase, there are three operations by a server, namely comparison, decrypt, drop, and XOR. So, in the process of authenticating the client in the mutual verification phase, the time complexity of the server is (T_ENC_ + T_XOR_). In the whole mutual verification phase, the time complexity of the tag is (Tc + 5T_ENC_ + 7T_XOR_). In the update phase, there is one operation by a client; the time complexity of the tag is (T_XOR_) in the update phase. Table 4 presents the time complexity of the proposed protocol.

The authors also performed their analysis on the space complexity. The client needs to store the hashed identity of the server and the client’s own identity. Thus, the space complexity of the client is 2 L. The case for the server side is similar. So, the space complexity of the reader is also 2 L. The comparison results are shown in Table 5.

The results obtained from both analyses show the suitability of our proposal in the current computing environment.

## 8. Conclusions and Future Work

A novel mechanism for mutual authentication based on hashing and PUF is proposed. It has been seen that previously proposed mechanisms have several security issues. The proposed mechanism utilizes the concept of PUF to protect the client from many physical attacks. PUFs are very strong mathematical computations based on a device’s manufacturing details and hence are impossible to break. Server verification, client verification, and mutual authentication processes are made more secure to make the mechanism resistant to impersonation, traceability, disclosure, desynchronization, and forward/backward secrecy. The proposed mechanism was also analyzed on the Scyther tool, where no attack within bounds was detected. The authors checked the status for all the secret IDs, random nonce values, and other functions used. All the statuses received from the Scyther tool were satisfactory. Hence, the authors can conclude that the proposed mechanism is more robust and secure than other available authentication mechanisms for a client-server scenario.

Modern application architecture relies heavily on APIs in today’s computing environment. Since API (security) was first incorporated into web-based applications, a great deal has changed. It is clear that API traffic has increased rapidly, some API protocols have gained traction, many new API security vendors and solutions have appeared, and, of course, attackers are developing new techniques and skills to compromise APIs. The need for authentication will remain the same between communicating parties, as previously, in order to make them mutually verified. Since integration among services requires low-complexity algorithms, the field of lightweight security design will remain an open topic for the research community. The authors will further try to reduce the computations inside the communicating parties to keep the security of the communicating devices intact. The challenge–response process of the implied PUF will further be reduced to gain another level of lightweight security system design.

## Figures and Tables

**Figure 1 sensors-23-06307-f001:**
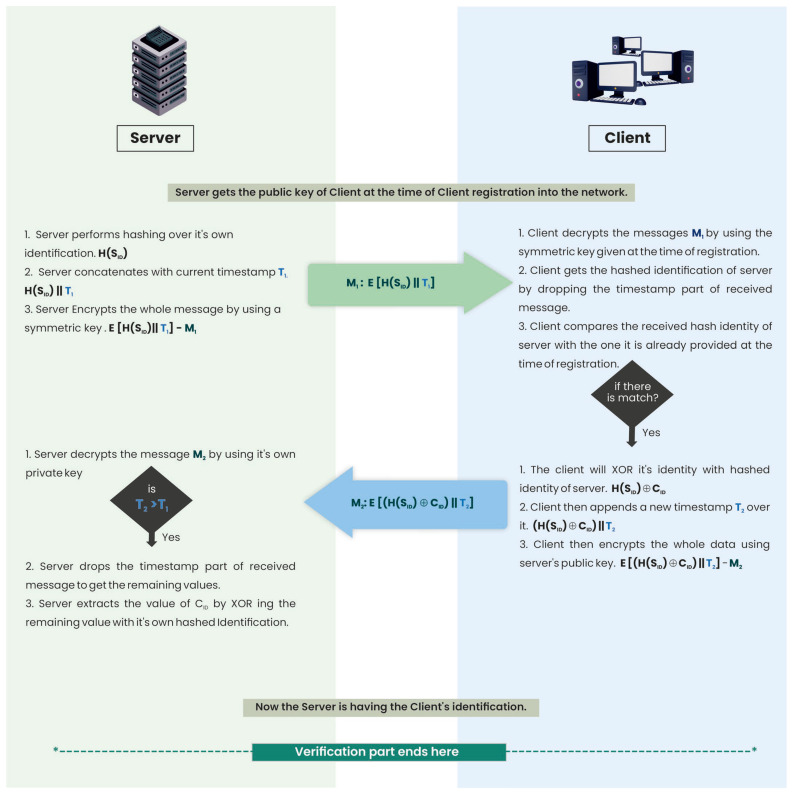
The proposed mutual authentication mechanism.

**Table 1 sensors-23-06307-t001:** Definitions of the symbols used in the proposed mechanism.

Symbol	Definition
C_ID_	Identification of client
S_ID_	Identification of server
H(S_ID_)	Hashing over identification of server
⊕	XOR operation
E	Encryption function
T	Timestamp
G_n_	Generator value
PUF	Physical unclonable function
F	Nonlinear-feedback shift register

**Table 2 sensors-23-06307-t002:** The proposed protocol verification results using the Scyther tool.

Claim				Status	Comments
Improved	S	improved, S1	Secret ID	OK	No attacks within bounds.
		improved, S2	Tiagree	OK	No attacks within bounds.
		improved, S3	Gnsynch	OK	No attacks within bounds.
		improved, S4	Alive	OK	No attacks within bounds.
		improved, S5	Weakagree	OK	No attacks within bounds.
	C	improved, C1	Secret SID	OK	No attacks within bounds.
		improved, C2	Tiagree	OK	No attacks within bounds.
		improved, C3	Gnsynch	OK	No attacks within bounds.
		improved, C4	Alive	OK	No attacks within bounds.
		improved, C5	Weakagree	OK	No attacks within bounds.

**Table 3 sensors-23-06307-t003:** The security comparison of the improved protocol to other protocols against advanced attacks.

Protocols	Impersonation	Traceability	Disclosure	Desynchronization	Forward/Backward Secrecy
Sadeghi et al. [31]	Non-resistant	Non-resistant	Resistant	Resistant	Non-resistant
Aysu et al. [46]	Resistant	Non-resistant	Non-resistant	Resistant	Non-resistant
Van Herrewege et al. [47]	Resistant	Resistant	Non-resistant	Resistant	Non-resistant
Kulseng et al. [40]	Resistant	Resistant	Non-resistant	Non-resistant	Non-resistant
Xu et al. [39]	Resistant	Resistant	Non-resistant	Non-resistant	Non-resistant
Bendavid et al. [38]	Resistant	Resistant	Resistant	Resistant	Non-resistant
Proposed	Resistant	Resistant	Resistant	Resistant	Resistant

**Table 4 sensors-23-06307-t004:** Result of time–cost analysis of the proposed protocol.

Protocol	Client Verification	Mutual Verification	Update
Kulseng [40]	-	2T_P_ + 3T_XOR_ + 2T_F_	T_XOR_ + T_F_
Xu et al. [39]	Tc	2T_P_ + 8T_XOR_ + 6T_LEFT_ + 2T_AND_ + T_OR_	2T_XOR_ + 6T_LEFT_ + 2T_AND_
Proposed	Tc	Tc + 5T_ENC_ + 7T_XOR_	T_XOR_

**Table 5 sensors-23-06307-t005:** Result of space–cost analysis of the proposed protocol.

Protocol	Client	Server
Kulseng [40]	3 L	4 L
Xu et al. [39]	3 L	9 L
Proposed	2 L	2 L

## Data Availability

No new data were created or analyzed in this study. Data sharing is not applicable to this article.

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
