# Peer review of "Hash and Physical Unclonable Function (PUF)-Based Mutual Authentication Mechanism"

_sensors, 2023, doi:10.3390/s23146307_

Round 1

Reviewer 1 Report

Dear authors, the problem addressed in the article is the vulnerability of web applications, particularly broken access control, which poses a significant security risk in enterprise environments. The existing mitigation strategies using cryptographic primitives, such as hashing and Physical Unclonable Function (PUF) have limitations in terms of practical integration, privacy preservation, and resistance to Denial of Service (DoS) attacks. The paper proposes a privacy-preserving authentication protocol that integrates the concepts of Hash and PUF in a practical manner. The protocol aims to mitigate the broken access control vulnerability while preserving the privacy of participating parties and providing resistance to DoS attacks. The proposed protocol overcomes the limitations of existing schemes and offers improved security properties. The analysis of performance demonstrates that the proposed solutions are secure, efficient, practical, and effective for web applications in an enterprise environment. However, this reviewer has several comments that must be addressed before it is considered for publication.

1) The paper contains numerous typographical errors. The authors should carefully proofread the paper to rectify these errors.

2) The utilization of the term 'our' is deemed inappropriate in a research paper due to its informal nature.

3) The excessive use of the pronoun 'we' throughout the paper should be minimized. Generally, 'we' is suitable for discussing future work in the conclusion, but aside from that, it should be used sparingly.

4) All figures appear blurry. The authors should either employ higher-resolution figures or recreate them as vector graphics.

5) The paper should be updated to incorporate more recent references, preferably from the past 2 or 3 years. The majority of references are older than 5 years. It seems the problem being tackled is not important/interesting to the research community anymore.

6) The authors should provide a clearer explanation of the contextual significance of this research, including why the research problem is important.

7) The contributions of the paper need to be highlighted more effectively. It should be explicitly stated what aspects are novel and how they address the limitations of previous work.

8) The authors should clearly delineate the differences between prior work and the solution presented in this paper.

9) The related work section lacks sufficient depth and should be expanded.

10) The authors should include a table that compares the key characteristics of prior work to emphasize their differences and limitations. Additionally, the table could contain a description of the proposed solution.

11) An example should be included by the authors to illustrate the problem definition.

12) It is advisable for the authors to provide an overview of their solution before delving into the details.

13) It is important to explicitly explain what aspects of the proposed solution are new and what aspects are not. In cases where parts are identical, appropriate citations should be used, and differences should be highlighted.

14) The description of the proposed solution should adopt a more formal tone.

15) The algorithms should be clearly described using pseudocode. The algorithms are described in a non-formal manner.

16) The complexity of the proposed solution should be discussed.

17) The experiments are too simple.

18) The experiments should be updated to include comparisons with more recent studies.

19) A statistical analysis should be conducted to demonstrate the significance of the experimental results.

20) There is insufficient discussion of the experimental results.

21) Additional text should be included to address future work or research opportunities.

1) The paper contains numerous typographical errors. The authors should carefully proofread the paper to rectify these errors.

2) The utilization of the term 'our' is deemed inappropriate in a research paper due to its informal nature.

3) The excessive use of the pronoun 'we' throughout the paper should be minimized. Generally, 'we' is suitable for discussing future work in the conclusion, but aside from that, it should be used sparingly.

Reviewer 2 Report

In general, this manuscript is meaningful. However, there are some problems to be solved and areas in need of more explanations.

 The topic could be relevant. There is new contribution from the paper. Some content is interesting. Please explain relevance to the focus of this journal “Sensors” more explicitly.

Please clarify the research problems, efforts and weaknesses of more related works, and to explain the necessity, innovation and comparative advantages of this paper more clearly.

The abstract mentioned 2021 project. Should 2023 status be updated by report or publications in 2023?

Could Figure 1 be better formatted so that there is no large space in page 6 around it ? Are all steps in the same level of abstraction?

The motivation and innovation are not clear. The authors spend too much space to discuss the basic concepts of PUFs and Hash function in Introduction, but ignore the security requirement in this field. They also mentioned that "Although there are many works has already been proposed using these mechanisms yet utilizing the concept of both simultaneously is not seen anywhere.", so the innovativeness and necessity of this paper seems to be a simple combination of PUF and Hash functions.

For the vulnerability of Kulseng's Mutual Authentication Protocol, the idea and effort of your proposed protocol should be further explained, before showing your protocol.

It lacks the experimental evaluation of the proposed protocol, and it would be better if there are experimental comparison with related protocols.

System model and threat model are required with technical details.

 Please explain the choice of the evaluation criteria, such as, explain with technical details why selected criteria are important for this paper’s objective. 

Explain in good technical details on complexity of the reviewed solutions.

Please discuss the weakness and limitation of this work, and the potential future works.

There are 40 references listed. Are all of them directly relevant to this paper’s research focus?

PUF in the title is not explained. Acronyms should be explained at first usage.

Reviewer 3 Report

The paper Hash and PUF based Mutual Authentication Mechanism refers to a privacy preserving authentication protocol that incorporates the privacy of participating parties and provides resistance to Denial of Service (DoS) attacks. My remarks are as follows:

·         It is not an original paper, as the authors claim. Similar considerations can be found in Secured Cloud Communication Using Lightweight Hash Authentication with PUF (DOI:10.32604/csse.2022.021129) or A PUF-based anonymous authentication protocol for wireless medical sensor networks (maybe not so related but still using PUF and hash function https: //doi.org/10.1007/s11276-022-03070-1)

·         Although the authors at the beginning of the work assure that their solution is resistant to DDoS, nowhere in the article is it clearly proven.

·         The work does not have a comparison of the efficiency of the proposed solution concerning other solutions of this type.

·         The quality of the paper and language needs to be improved.

Must be improved.

Round 2

Reviewer 1 Report

The authors have addressed my comments and questions. I'm satisfied with them.

The quality of the text has been improved.

Reviewer 3 Report

All my concerns were properly addressed.